# Vulnerability Analysis to Drought Based on Remote Sensing Indexes

**DOI:** 10.3390/ijerph17207660

**Published:** 2020-10-20

**Authors:** Huicong Jia, Fang Chen, Jing Zhang, Enyu Du

**Affiliations:** 1State Key Laboratory of Earth Surface Processes and Resource Ecology, Beijing Normal University, No.19, Xinjiekouwai St, Haidian District, Beijing 100875, China; jiahc@radi.ac.cn; 2Key Laboratory of Digital Earth Science, Aerospace Information Research Institute, Chinese Academy of Sciences, No. 9 Dengzhuang South Road, Beijing 100094, China; 3College of Resources and Environment, University of Chinese Academy of Sciences, Beijing 100049, China; duenyu@cug.edu.cn; 4Department of Geography, Beijing Normal University, No.19, Xinjiekouwai St, Haidian District, Beijing 100875, China; zhangjing_bnu163@163.com

**Keywords:** remote sensing index for drought, vulnerability, drought at risk populations, rapid assessment, the middle and lower reaches of the Yangtze River, China

## Abstract

A vulnerability curve is an important tool for the rapid assessment of drought losses, and it can provide a scientific basis for drought risk prevention and post-disaster relief. Those populations with difficulty in accessing drinking water because of drought (hereon “drought at risk populations”, abbreviated as DRP) were selected as the target of the analysis, which examined factors contributing to their risk status. Here, after the standardization of disaster data from the middle and lower reaches of the Yangtze River in 2013, the parameter estimation method was used to determine the probability distribution of drought perturbations data. The results showed that, at the significant level of α = 0.05, the DRP followed the Weibull distribution, whose parameters were optimal. According to the statistical characteristics of the probability density function and cumulative distribution function, the bulk of the standardized DRP is concentrated in the range of 0 to 0.2, with a cumulative probability of about 75%, of which 17% is the cumulative probability from 0.2 to 0.4, and that greater than 0.4 amounts to only 8%. From the perspective of the vulnerability curve, when the variance ratio of the normalized vegetation index (NDVI) is between 0.65 and 0.85, the DRP will increase at a faster rate; when it is greater than 0.85, the growth rate of DRP will be relatively slow, and the disaster losses will stabilize. When the variance ratio of the enhanced vegetation index (EVI) is between 0.5 and 0.85, the growth rate of DRP accelerates, but when it is greater than 0.85, the disaster losses tend to stabilize. By comparing the coefficient of determination (R^2^) values fitted for the vulnerability curve, in the same situation, EVI is more suitable to indicate drought vulnerability than NDVI for estimating the DRP.

## 1. Introduction

Drought, one of the major natural hazards worldwide, causes water shortages that not only increase the vulnerability of the agricultural sector and its economic losses but also greatly threatens human life [1,2,3]. China’s economic losses caused by meteorological disasters accounted for 71% of all of its natural disaster losses from 1949, including 53% caused by drought disasters [4]. In recent decades, due to the increasing global warming, the frequency and intensity of drought events have also trended upward.

In researching drought, most scholars tend to focus on climatological characteristics, namely hazard-inducing mechanisms and risk assessment [5,6,7,8], especially in arid and semi-arid areas [9,10]. For this work, the research objects are mainly agricultural systems and crops [11,12], and methods used often involve statistical analysis, model evaluation, remote sensing monitoring, and field survey, among others [13,14,15,16]. To date, the research on drought vulnerability generally relies on two key approaches: qualitative analysis and quantitative analysis. Qualitative analysis mainly studies the formation mechanism(s), change trend(s), influencing factors, and geographical locality impacted by drought disturbances [17,18]. Quantitative analysis primarily tries to quantify the drought system’s elements, usually by building a vulnerability assessment index system and then evaluating the vulnerability of the agricultural drought via a comprehensive assessment model [19]. Nevertheless, to minimize disaster losses, the timely warning of natural disasters or initiation of emergency responses to them is essential; hence, in carrying out such research, it is often necessary to complete the assessment of possible disaster losses as quickly as possible. In the case of small data sets or incomplete data, preliminary judgments can help to organize and implement post-disaster relief work in advance.

Previous studies have focused on the risk of drought in northern areas of China. The rate of change in meteorological droughts has risen from 1961 to 2019 in China [20], whose drought-prone area has shifted from the two zonal distribution belts of its southwest and north China in the 1980s to the middle reaches of the Yangtze River and Yellow River [21,22]. Due to more frequent extreme climate events, precipitation in China’s southern region is undergoing a slight increase as temperature rises significantly [23]. The middle and lower reaches of the Yangtze River feature modernized agriculture and densely populated areas, being among the most economically developed regions in China, one located in subtropical monsoon zone, with sufficient rainfall but uneven distribution, where droughts and floods are frequent [24]. As a representative water shortage area, economic production and social lifestyle of the area depend on the availability of abundant water resources. Nevertheless, drought prevention and resistance infrastructure is relatively lacking there, making the impact of drought more serious. In the rainy areas of southern and eastern China, droughts have increased in the past 10 years [20]. Therefore, studying the formation mechanisms of drought in the middle and lower reaches of the Yangtze River is of great practical significance.

In the summer of 2013, the high temperatures in the middle and lower reaches of the Yangtze River were significant, and precipitation there was abnormally low, in fact at its lowest within the past 60 years. The water level in the main channel of the middle and lower reaches of the Yangtze River was at its lowest or close to the lowest point in the same period (June to August) in history [25]. This summer, drought inevitably limited the drinking water accessible to people and animals in the geographical area incurring drought. The direct economic loss in Hunan Province alone reached as high as 14.3 billion Yuan [26]. Many scholars have studied the causes of summer precipitation anomalies in the middle and lower reaches of the Yangtze River from multiple perspectives, such as atmospheric circulation anomalies and the effects of external forcing [27,28,29]. In recent years, the rapid assessment of disaster losses has gradually become the key task of disaster assessment research, and some scholars have made certain progress. Given the characteristics of drought disasters, the remote sensing index for drought was selected as their hazard-inducing factor. Specifically, because such an index has the characteristics of wide coverage, strong real-time information collection, easy processing, and good continuity, it is deemed more suitable for rapid loss assessment. In order to link with the “Statistical System of Natural Disasters“ [30], those populations with difficulties in accessing drinking water because of drought (hereon “drought at risk populations”, abbreviated as DRP) were selected to reflect the overall drought disaster situation, for which a drought vulnerability curve was constructed to express the quantitative functional relationship between hazard and loss. A vulnerability curve is an important tool for the rapid assessment of drought losses, providing a scientific basis for both drought risk prevention and post-disaster relief interventions.

This study had three objectives: (1) to use a remote sensing index for drought to monitor the temporal and spatial pattern of drought; (2) to analyze the distribution of drought-related hazards and disasters from the perspective of probability; (3) to obtain the vulnerability curve fitted based on remote sensing drought index and disaster losses, thus providing a basis for rapid disaster assessment.

## 2. Materials and Methods

### 2.1. Study Area

The middle and lower reaches of the Yangtze River, with low and flat terrain, interlaced river networks, and numerous lakes, whose surface structure is shaped by low mountains and alternating hills and plains. It is one of the regions distinguished by an abnormal rainfall regime and frequent droughts and floods in China. Geographically, the middle and lower reaches of the Yangtze River corresponds to a vast area north of the Nanling Range, south of the Qinling Mountains–Huaihe River Line, and east of Wu Mountain, thus lying 110°E longitudinally spanning 25°~34°N in latitude. Here, the average annual temperature is 14 °C~18 °C, and the annual precipitation is 1000~1400 mm, falling mostly in spring and summer [31]. In this region is one of the three major plains of China, with abundant water resources and a high land reclamation index, being an important production base for the country’s grain, cotton, and oil, as well as a crucial industrial base. Considering those provinces incurring the severe drought in southern China during the summer of 2013, Hunan, Hubei, Zhejiang, Jiangxi, Anhui, Chongqing, and Guizhou were selected as the study area (Figure 1).

### 2.2. Materials

Given the current demand for drought vulnerability analysis based on past disasters, this paper collected and organized time-series data sets to achieve our research objectives (Table 1). This historical disaster data set was mainly derived from the daily report of disaster data, provided by the National Disaster Reduction Center of the Ministry of Emergency Management, for the summer of 2013 during that year’s pronounced southern drought event (615 counties, lasting from 9 July to 18 September). Taking into account the completeness and continuity of this data, this study selected the DRP as the object of the drought vulnerability study. Next, the drought remote sensing index corresponding to each disaster process has been selected as the hazard inducing factor, and a total of 24 scenes from July to September 2013 have been downloaded from the NASA official website (https://ladsweb.modaps.eosdis.nasa.gov). MOD13Q1 is the synthetic data of MODIS Global Vegetation Index, which includes two vegetation indexes: NDVI (normalized vegetation index) and EVI (enhanced vegetation index). Meteorological data came from daily ground observation data collected at 90 meteorological observation stations in the middle and lower reaches of the Yangtze River, provided by the China Meteorological Administration, from January 1981 to December 2013, which included 24-h precipitation and daily average temperature. Basic geographic data, such as administrative divisions and rivers, was obtained from the National Geomatics Center of China.

### 2.3. Methods

First, the county-level disaster data from the national disaster management system were sorted according to disaster events; that is, a record must match the initial report, renewal, and verification to be considered as valid sample in the data. Using MRT (MODIS Reprojection Tool, Windows version, LP DAAC - Land Processes Distributed Active Archive Center, Sioux Falls, USA), the image was re-projected; then, we combined and cut out the images of the same date with the vector file of the study area, to obtain the NDVI and EVI drought indices data of different dates in the study area. Second, the DRP was selected as the hazard-inducing factor for drought vulnerability analysis. In MINITAB software (version 17.1, Minitab, LLC, State College, PA, USA), the probability map method was used to estimate the optimal parameters of the disaster sample data, so as to determine the probability distribution form of the disaster. Statistical analyses of the distribution of drought loss data were made using probability density and cumulative probability functions. Finally, combining those with NDVI and EVI drought-indices data, the vulnerability curve analysis of the DRP was implemented.

#### 2.3.1. Remote Sensing Indices for Drought

More than 40 vegetation indices have been proposed, among which the NDVI index is currently the most widely used [32,33]. The NDVI was first proposed in 1973, by Rouse et al. [34]; its calculation for a given pixel always results in a number that ranges from minus one (−1) to plus one (+1). In the use of remote sensing imagery for vegetation research and plant phenology research, NDVI is broadly used to convey information on vegetation growth [35,36]. Since NDVI can eliminate most of the changes in irradiance related to the atmosphere, it can strengthen the response ability of terrestrial vegetation to the PAR (photosythentically active radiation) spectrum. The NDVI is calculated using the near-infrared (NIR) and visible red bands of a multispectral sensor [37], with this formula:(1)$$\mathrm{NDVI}=\frac{\mathrm{NIR}-\mathrm{RED}}{\mathrm{NIR}+\mathrm{RED}}$$
where NIR is the near infrared band value for a cell, and RED is the red band value for that cell. (NDVI can be calculated for any image that has a red and a near infrared band.) The biophysical interpretation of NDVI is the fraction of absorbed PAR. Specifically, EVI improves on NDVI’s spatial resolution, being more sensitive to differences in heavily vegetated areas, and it better corrects for atmospheric haze as well as the land surface beneath the vegetation [38,39]. The EVI was specifically developed to be more sensitive to changes in areas having high biomass (a serious shortcoming of NDVI), to reduce the influence of atmospheric conditions on vegetation index values and to correct for canopy background signals [40]. EVI can be associated with stress and vegetation changes related to drought events.
(2)$$\mathrm{EVI}=2.5\times \frac{\left(\mathrm{NIR}-\mathrm{RED}\right)}{\left(\mathrm{NIR}+C_{1}\times \mathrm{RED}-C_{2}\times \mathrm{BLUE}+L\right)}$$
where NIR, RED, and BLUE are atmospherically corrected (or partially atmospherically corrected) surface reflectance values, and C1, C2, and L are coefficients to correct for the atmospheric condition (i.e., aerosol resistance). For the standard MODIS EVI product: L = 1, C1 = 6, and C2 = 7.5.

A statistical analysis of the pixels included in the study area is performed, and the average value of NDVI and EVI for each pixel is obtained. For change detection, the Formula (3) is used to calculate the difference between NDVI_i_ and NDVI_1_ in each scene image; where $V_{\mathrm{NDVIj}}$ is the Variance Ratio of NDVI_j._ NDVI_1_ is the initial NDVI value on 9 July 2013. The Formula (4) is calculated in the same way with the Formula (3).
(3)$$V_{\mathrm{NDVIi}}=\frac{\left({\mathrm{NDVI}}_{1}–{\;\mathrm{NDVI}}_{i}\right)}{{\mathrm{NDVI}}_{1}}\times 100\%$$
(4)$$V_{\mathrm{EVIi}}=\frac{\left({\mathrm{EVI}}_{1}–{\;\mathrm{EVI}}_{i}\right)}{{\mathrm{EVI}}_{1}}\times 100\%$$

#### 2.3.2. A probability Analysis Method

In probability theory, the probability density function (PDF) is the probability function describing the density of a continuous random variable lying within a range of certain values [41,42]. The probability of the random variable falling within a particular range of values is given by the integral of this variable’s density over that range—that is, it is given by the area under the density function, but above the horizontal axis, and between the lowest and greatest values of the range. It is denoted by f(x). This function is either positive or non-negative at any point of the graph and the integral of PDF over the entire space is always equal to one (i.e., sums to unity). In the case of a continuous random variable, the probability taken by X at some given value x is always 0. In this case, if we find P(X = x), it does not work. Instead of this, we must instead calculate the probability of X lying in a defined interval (a, b), for which we may then calculate it for P (a< X< b). This pivotal step can be accomplished through a PDF.

In cumulative frequency analysis, the frequency at which values of a phenomenon less than a reference value occur is analyzed mathematically [43,44]. Such a cumulative frequency is also called the frequency of non-exceedance probability. In a set of data or observations, cumulative frequency may be used to determine the number of observations, which lie above a specific observation. To calculate the cumulative frequency for such a particular observation, the frequency of that observation is added to the sum of frequencies of its predecessors. The last observation’s cumulative frequency value is the sum of all the frequencies of the entire data set.

#### 2.3.3. A Vulnerability Curve

Not all disaster history events have data records. When the index method is not sufficiently standardized and the evaluation results are not entirely credible, the vulnerability curve provides a new way of evaluating vulnerability to drought. A vulnerability curve is also called a “vulnerability function” or “disaster rate function” or “disaster rate curve” [45,46]. It first appeared in 1964, and since then, it has been used to measure various disasters of different intensities and their corresponding losses (rates). The relationship between them is mainly expressed in the form of curves, surfaces, or tables. For example, based on a crop model, the vulnerability curve between crop water stress and corn yield reduction rate was simulated, and the drought risk of maize in China was accordingly evaluated [47]. The vulnerability curve truly faces the individual disaster-affected body (for example, population, crop, infrastructure and building) and can fundamentally solve the characteristics of the vulnerability evaluation results [15,18,47], such as poor pertinence or weak operability, in addition to capturing the overall vulnerability characteristics of the region via the disaster-affected body’s own vulnerability. Therefore, research on the “vulnerability curve” can provide new ideas and methods for conducting in-depth research on drought risks. By fitting the function between drought hazard intensity (y-axis) and drought loss rate (x-axis), a vulnerability curve could thus be attained.

While setting exposure to 1 (disaster-affected regions), the risk of each assessment cell was a function of hazard intensity and vulnerability. The formula *R = f (H, V, E)* is a theoretic equation. *H* is indicated through the hazard intensity-probability curve. *V* is indicated through the hazard intensity-loss rate curve. *E (exposure)* is assigned to 1 if it is in disaster-affected regions, setting E to 0 if not. The loss rate *(L)* under certain hazard intensity delegates the value of *V*. Drought disaster risk is the loss rate under a certain level of hazard. Natural disaster risk is the loss rate under a certain level of hazard (Figure 2).

## 3. Results

### 3.1. Spatiotemporal Pattern of Drought

#### 3.1.1. Temporal Distribution Characteristics

According to statistics from the meteorological stations, in the 2013 summer, the average temperature of the seven provinces (cities) in the middle and lower reaches of the Yangtze River was 28 °C, 1.6 °C higher than normal, and the highest since 1961 (Figure 3). The average maximum temperature in some areas reached 36 °C~40 °C. The highest temperatures in parts of north-central Zhejiang, Shanghai, southern Jiangsu, southeastern Anhui, northwestern Jiangxi, and northern Hunan were between 40 °C and 44 °C. The average rainfall in this study area is only 398.4 mm, 18.0% less than normal (486.4 mm), and the lowest since 1993. Among them, the northeastern part of Guizhou province and the western part of Hunan province received 50% to 80% less precipitation, with rainfall totaling less 200 mm over three months in the study region. In Hengyang City, Hunan Province, there was not a drop of rain from July to early August, a record drought and the longest in 63 years.

#### 3.1.2. Spatial Distribution Characteristics

Severely low precipitation and continued high temperatures have caused droughts and impacted development in parts of China south of the Yangtze River and in the eastern part of southwest China. According to NDVI and EVI monitoring, light to moderate drought events occurred in parts of southwestern and northeastern Guizhou and Hunan Provinces as well as in north-central Jiangxi, in mid-to-late June 2013. These droughts developed in mid-to-late July. By the end of July, moderate to severe droughts prevailed in most of Guizhou and Hunan Province, in addition to central and eastern Chongqing City. Extreme drought occurred in some areas of Zunyi City, Guizhou Province, whereas light to moderate droughts happened in parts of north-central Zhejiang Province, central Jiangxi Province, and central Yunnan Province. Generally, the summer’s drought continued to develop through the first 10 days of August. Meteorological droughts of moderate or above levels characterized most parts south of the Yangtze River, with severe to extreme droughts occurring in most parts of Guizhou and Hunan Province. In mid-August, the extreme drought range was reduced, but severe droughts began to occur and expand in western and eastern Jiangxi Province, central and southern Zhejiang Province; hence, regionally, the drought was still very serious. In late August, the drought incurred in the south gradually eased, going from east to west (Figure 4 and Figure 5). The normalized variance ratios were used in Figure 4 and Figure 5.

To further reveal the mechanistic drivers of drought-causing disasters, the PDF and cumulative distribution function (CDF) were used here to derive statistics on the relevant feature quantities and trend distribution of the variance ratios of the NDVI and EVI indicators. The vast majority of NDVI’s variance ratio was concentrated at 0.6–0.85, having a cumulative probability of about 78%, of which the cumulative probability of the 0.4–0.6 range was 12%, and the cumulative probability for that above 0.85 was 9% (Figure 6 and Figure 7). The bulk of EVI’s variance ratio lays between 0.4 and 0.6, for which the cumulative proportion was about 77%. By contrast, the cumulative probability was 17% over the 0.2–0.4 range while that for values greater than 0.6 was just 6% (Figure 8 and Figure 9).

### 3.2. Spatiotemporal Analysis of Drought Disasters

From June to August 2013, the precipitation in Guizhou Province, Hunan Province, Jiangxi Province, Chongqing City, southern Hubei Province, southern Zhejiang Province, and northern Anhui Province continued to remain low, leading to continuous droughts in most areas of these seven provinces (cities). During its peak period, the drought adversely affected 79.764 million people, of whom 15.788 million had drinking water difficulties, and 12.743 million people needed living assistance. The affected area of crops totaled 84,140 km^2^, of which 12,072 km^2^ yielded no harvest. Further, 4.378 million large livestock also suffered from a lack of drinking water. In all, the direct economic loss from this regional drought was a staggering 50.83 billion Yuan [25].

#### 3.2.1. Temporal Distribution Characteristics

By using the reported disaster data from the 9 July through 2 September 2013, the temporal distribution characteristics of the main disaster indicators, such as the drought at-risk populations and the population in need of assistance in the study area (provided by the Ministry of Emergency Management of China), were analyzed (Figure 10). The disaster situation in July was generally mild and any changes at this time were slight, being rather stable overall. However, the disaster situation quickly advanced from the end of July through the start of August, peaking later in mid-August. In late August, due to the impact of the rains brought in by southwest monsoon and typhoon “Utor”, the drought situation was relieved to a certain extent, and the disaster situation declined. Nevertheless, because of the uneven distribution of precipitation across time and space, the drought in western Guizhou had not been completely alleviated, so its disaster situation increased slightly. After entering September, with the continuous autumn rain that fell, the drought over the entire region ended.

As Figure 10 shows, a relatively pronounced moderate meteorological drought occurred on 9 July, with its government-issued status occurring about 20 days later, on 30 July; that is, the first peak of the DRP and those in need of assistance because of drought impacts. Later, on 12 August, the second disaster peak was evident. Therefore, the time lag between socioeconomic drought impact and the actual meteorological drought event is about 20 days.

#### 3.2.2. Spatial Distribution Characteristics

Generally, in terms of the spatial extent of drought, the 2013 summer drought was mainly distributed in central and western Hunan Province, central and northern Hubei Province, central and western Jiangxi Province, western Guizhou Province, and southern Chongqing City (Figure 11).

Among them, those that suffered heavily were Hengyang City, Yongzhou City, Yiyang City, Shaoyang City, and Loudi City in Hunan Province; Xiaogan City, Xiangyang City, and Suizhou City in Hubei Province; Shangrao City and Yichun City in Jiangxi Province; Zunyi City and Tongren City in Guizhou Province; and Bijie City and some other cities, whose respective populations with drinking water difficulties exceeded 300,000. Worse still, more than 600,000 people in Hengyang, Xiaogan, and Zunyi encountered difficulty in accessing drinking water.

#### 3.2.3. Disaster Loss Analysis

To make the drought disaster data of different counties comparable, the historical disaster data was first standardized. The parameter estimation method was used to determine the PDF of the overall drought-related disaster, by calculating the unknown parameters via OLS (ordinary least squares) contained in the overall distribution based on the sample data drawn from the disaster situation. It is the premise of analyzing or inferring the essential pattern of the disaster situation data. Using the MINITAB software (version 17.1) tool, the probability plot method let us estimate the optimal parameters of the disaster sample data. The candidate parameter models had four distribution forms: Lognormal, Exponential, Gamma, and Weibull. Through the estimation and analysis of the optimal parameters of the probability, the estimated parameters and probability distribution plot of the DRP were obtained (Figure 12). According to the Anderson-Darling (AD) goodness-of-fit statistic and its associated *p*-value [48], the results show DRP is significantly passing the hypothesis testing (at α = 0.05) and followed the Weibull distribution, for which the best-fitting parameters obtained (i.e., *p*-value is greater than 0.05; the smaller the AD value, the better the fit).

It can be seen from the probability density and cumulative probability distribution in Figure 13 and Figure 14, respectively, that most of the losses (DRP) are concentrated at 0~0.2, for which the cumulative probability is about 75%. The cumulative probability of 0.2 to 0.4 is 17%, and that of greater than 0.4 is just 8%. From the statistical characteristics of the PDF and CDF of the affected populations, evidently, drought has a certain impact on the populations, such that more than 90% of drought events caused the proportion of the affected populations to be below 0.4.

### 3.3. Vulnerability Analysis

To further determine the magnitude of the 2013 drought disaster under different disaster intensities, the drought remote sensing index was selected as a proxy for hazard intensity, and the vulnerability curve of the DRP corresponding to different remote sensing indexes for drought was fitted, to construct a functional relationship between the hazard intensity and the loss. On the basis of observing the distribution mode of historical case sample data, the Logistic growth curve was selected here, as the basic line type of the vulnerability function, with 95% confidence intervals, and the convergence tolerance of less than 0.01. The Gauss–Newton algorithm was used for multiple iterations. The vulnerability curves for NDVI–DRP and EVI–DRP were obtained with these formulae:(5)$$y_{1}=1-\exp(-\exp(-11.9+14.8x1))\;\;\;\;\;\;\;\;\;r^{2}\;=\;0.59$$

In the above formula, *y1* is the standardized DRP, and *x1* is the variance ratio of NDVI.
(6)$$y_{1}=1-\exp(-\exp(-5.89+8.18x2))\;\;\;\;\;\;\;\;\;r^{2}\;=\;0.61$$

In the above formula, *y1* is the standardized DRP, and *x2* is the variance ratio of EVI.

As seen in the vulnerability curves (Figure 15 and Figure 16), as the variance ratio of the drought index continues to increase, the affected population also rises. When the variance ratio of NDVI is between 0.65 and 0.85, the growth rate of standardized DRP is relatively fast, but when it exceeds 0.85, this growth rate is relatively slow, and the disaster loss tends to be stable. When the variance ratio of EVI is between 0.5 and 0.85, the growth rate of standardized DRP accelerates most, but tends to stabilize at values greater than 0.85. In comparing the coefficient of determination (R^2^) between the two vulnerability curves, under the same circumstances, the EVI had better goodness-of-fit, suggesting its accuracy for predicting the DRP is better than that based on NDVI.

## 4. Discussion

Many scholars monitor drought disasters based on one-year’s worth of remote sensing data, and they have achieved some noteworthy results by implementing and evaluating mathematical models to such data. Nevertheless, their practical value still lags somewhat, as they cannot fully meet the needs of the agricultural production sector and ensure the safety of people’s lives. The data analyzed correspond to the year 2013 as there is a severe drought episode in the study area. We used just data for 2013 to evaluate drought; historical data (>20 years) may need to be used as drought is a climatic disaster.

The most well-established approach to agricultural drought monitoring is the NDVI [13,35,36,37]. In the future, we should strive to carry out multiple or many years of continuous similar monitoring based on remote sensing data. In particular, more attention should be paid to the comparison and analysis of various drought indices, by seeking the best drought indicators that can clearly convey the temporal evolution and spatial differentiation of drought stress at various relevant scales, so as to improve the accuracy of monitoring drought events and their impacts. For a more fair analysis, more studies should be conducted based on various drought indices, which integrated vegetation indices (NDVI and EVI) along with the surface temperature and evapotranspiration information, such as the Vegetation Condition Index (VCI) [38], the Temperature Condition Index (TCI) [39] and the Temperature Vegetation Drought Index (TVDI) [10].

Different evaluation indicators and a variety of established models to evaluate vulnerability have been constructed [49,50]. By contrast, quantitative research on drought vulnerability is still in its exploratory stage. The established drought vulnerability measurement indicators are mostly rooted in hydrology, meteorology, and geography, with far less consideration given to social and economic indicators, so any selected index used alone has certain inherent limitations. Our linear regression results for disaster loss and drought index show that under the interference of multiple factors such as human activities, there is no single, reliable socio-economic index that can capture the socio-economic impact caused by drought events.

Similarly, the drought risk assessment of southwest China has been conducted based on a crop model [18]. The probability density of the drought hazard index is mainly located between 0.2 and 0.6. The exceeding probability curves show that the steepest curve is at 0.35 [18,47]. The final vulnerability curves and corresponding functional equations are all determined by a nonlinear regression model using MATLAB software. Artificial neural networks (ANNs) have proven to be successful in dealing with complicated problems, such as function approximation and pattern recognition [51]. Due to their powerful capability and functionality, ANNs provide an alternative approach for many assessment problems that are difficult to solve by conventional approaches [52]. The Backpropagation (BP) neural network is currently the most widely used ANN [6]. Further studies using a BP neural network for rapid assessment of drought at-risk populations are still required.

The drought vulnerability assessment method in this paper is mainly based on the probability and statistics method of random theory. That is, we used mathematical statistics to analyze and refine past disaster data, to elucidate the regularities of disaster development and to calculate the risk probability needed for achieving the purpose of predicting and evaluating future disaster risks. The probability statistical method based on the drought loss index assumes the drought loss data constitutes a random variable [53], but in fact, drought loss is often the outcome of human subjective interventions and so it does not conform to random behavior per se. The basic idea behind the drought risk assessment method based on the process of drought events is to first identify all drought events through hydro-meteorological elements and then establish a quantitative relationship between drought frequency and potential loss/drought resistance capacities to better describe the drought risk faced by society. This kind of method is based on the physical process of drought risk formation, and it can incorporate internal connections and the evolution process between multiple key risk components.

Starting from the mechanism of the drought occurrence process, it is generally believed that the components of drought risk should include drought hazard (H, hazard), regional exposure (E, exposure), and environmental vulnerability (V, vulnerability) [54,55]. In addition, the strength of drought resistance behavior in drought-affected areas will also affect the amount of drought losses. This subjective element is called drought resistance (RE, resistance). For areas with similar climate and socio-economic conditions, comprehensive indicators can be used to evaluate the relative magnitude of drought risk, which could be of guiding significance for improved drought risk management.

## 5. Conclusions

Currently, studies related to climate change are a source of research and necessary to know its different effects, in this case to evaluate the scope and improve the prevention and action of the risk of drought. A vulnerability assessment can provide a scientific basis for drought risk prevention and post-disaster relief. This study focused on impact both for its relatively unusual nature (i.e., drinking water rather than agricultural loss) as well as its pragmatic purpose. The key conclusions are as follows:(1)The “drought at risk populations” (DRP) is used as a factor for the vulnerability analysis of a drought disaster in 2013. According to the Anderson–Darling goodness-of-fit statistic and the associated *p*-value, DRP is best described by the Weibull distribution.(2)According to the probability density function and cumulative distribution function of the affected population, the proportion of the latter was below 0.4 for more than 90% of the drought events. Most of the losses (DRP) are concentrated in the 0~0.2 range, whose cumulative probability is about 75%.(3)Research on the “vulnerability curve” can provide new insight and approaches when pursuing in-depth research on drought risk. Through non-linear line fitting of a function between drought hazard intensity and drought loss rate, we demonstrated that a vulnerability curve is realizable. According to its distributions, the affected population increased in response to increasing variance ratio if drought index continues to increase. Under the same circumstances, when compared with NDVI, the vulnerability of curve of EVI provides a better fit (R^2^) and hence accuracy for estimating DRP.

## Figures and Tables

**Figure 1 ijerph-17-07660-f001:**
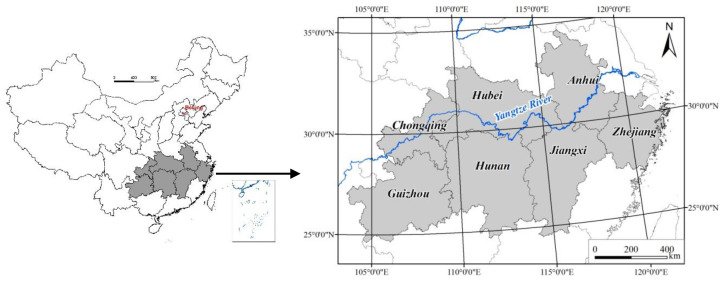
Location of the study area along the Yangtze River, China.

**Figure 2 ijerph-17-07660-f002:**
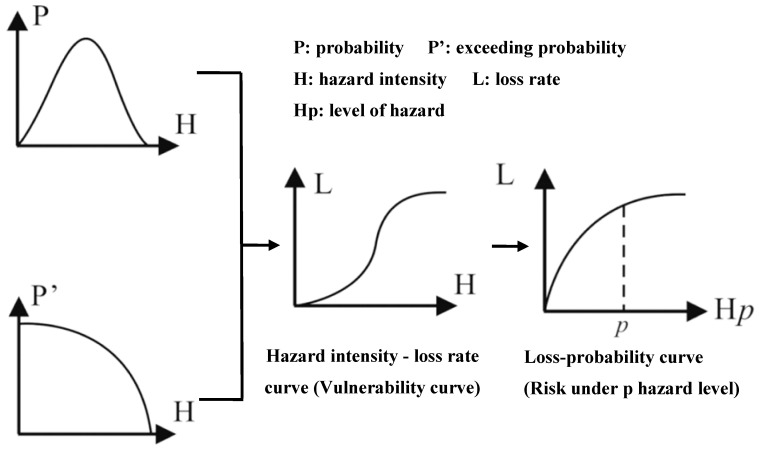
Sketch of assessment of natural disaster risk.

**Figure 3 ijerph-17-07660-f003:**
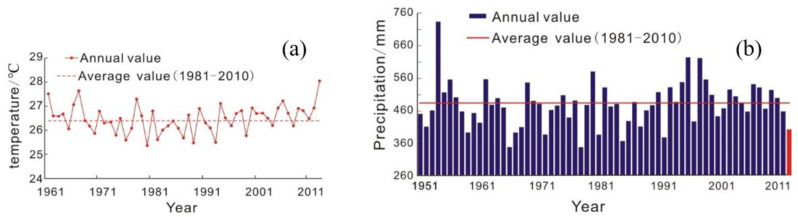
Changes in average summer temperature and precipitation over the last 50–60 years. (**a**) Temperature; (**b**) precipitation.

**Figure 4 ijerph-17-07660-f004:**
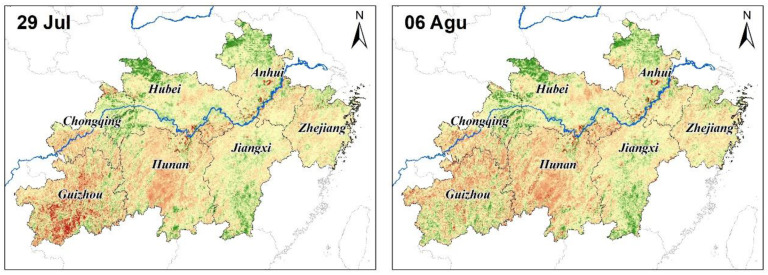
Time series of variance ratio of the normalized vegetation index (NDVI) in the middle and lower reaches of the Yangtze River, China.

**Figure 5 ijerph-17-07660-f005:**
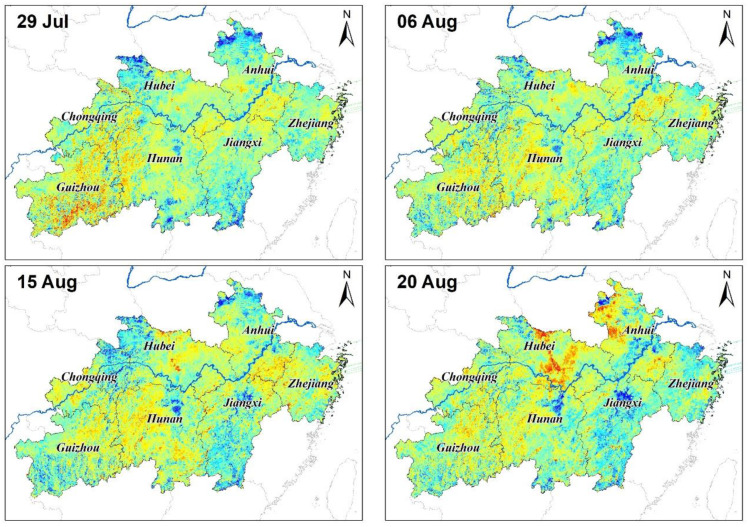
Time series of variance ratio of the enhanced vegetation index (EVI) in the middle and lower reaches of the Yangtze River.

**Figure 6 ijerph-17-07660-f006:**
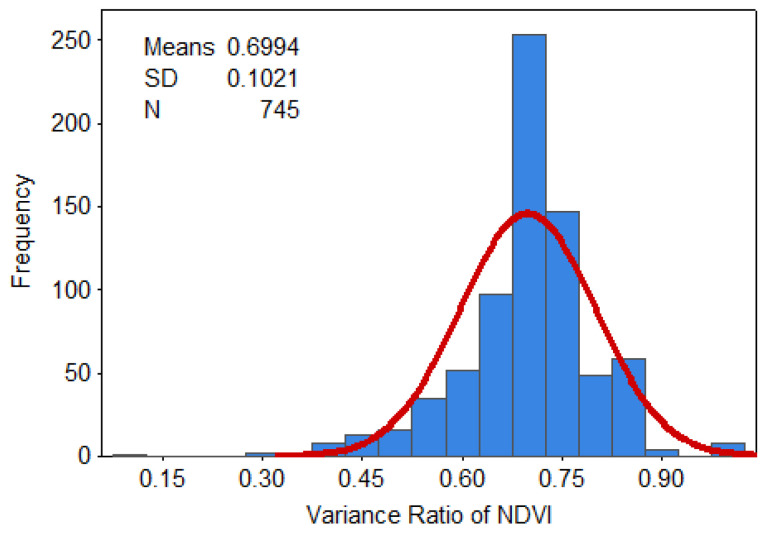
Probability density for the variance ratio of NDVI.

**Figure 7 ijerph-17-07660-f007:**
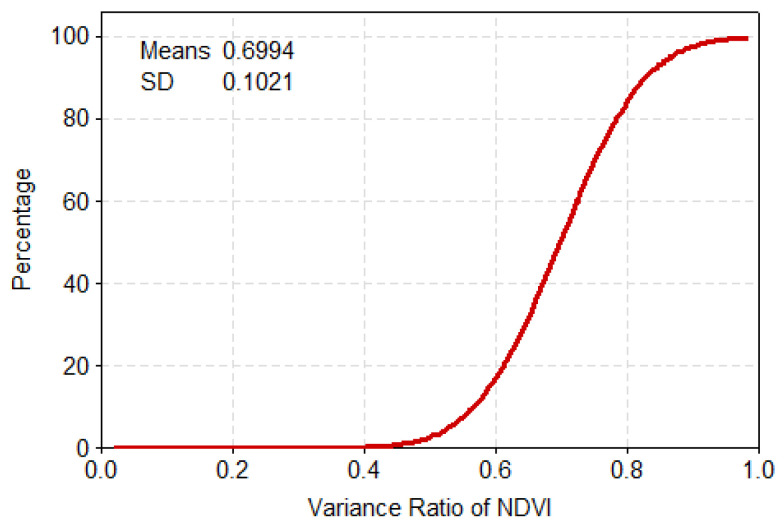
Cumulative probability for the variance ratio of NDVI.

**Figure 8 ijerph-17-07660-f008:**
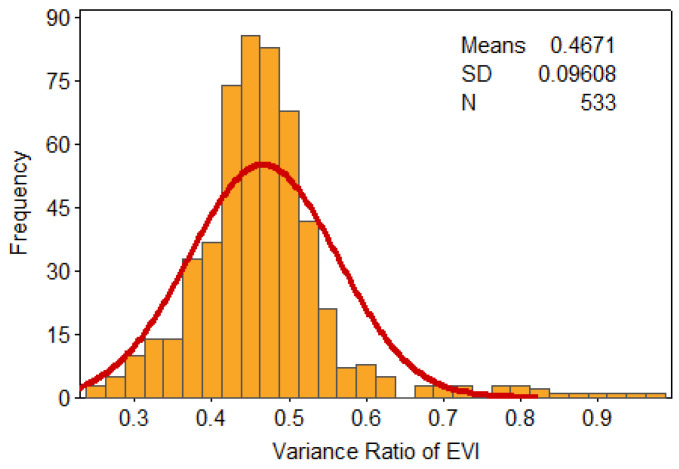
Probability density for the variance ratio of EVI.

**Figure 9 ijerph-17-07660-f009:**
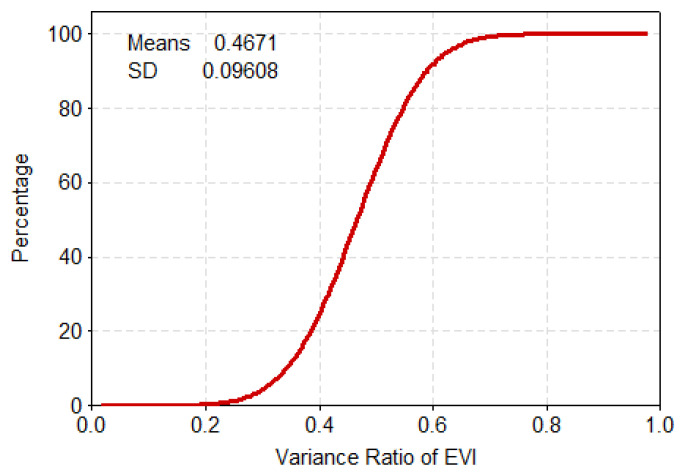
Cumulative probability for the variance ratio of EVI.

**Figure 10 ijerph-17-07660-f010:**
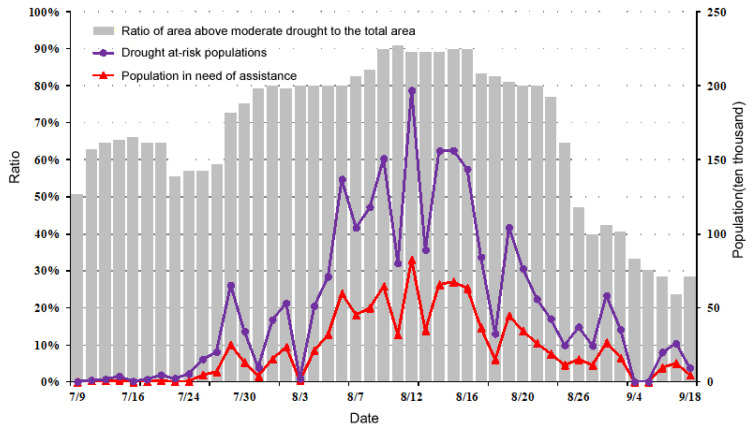
Time series change of the drought at risk populations (DRP) during summer drought in the middle and lower Yangtze River in 2013.

**Figure 11 ijerph-17-07660-f011:**
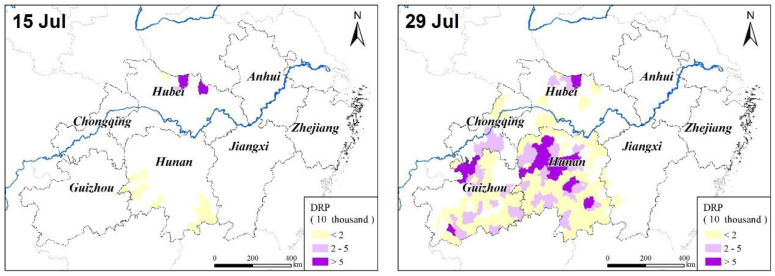
The spatial distributions of the DRP during the 2013 summer drought in the middle and lower reaches of the Yangtze River, China.

**Figure 12 ijerph-17-07660-f012:**
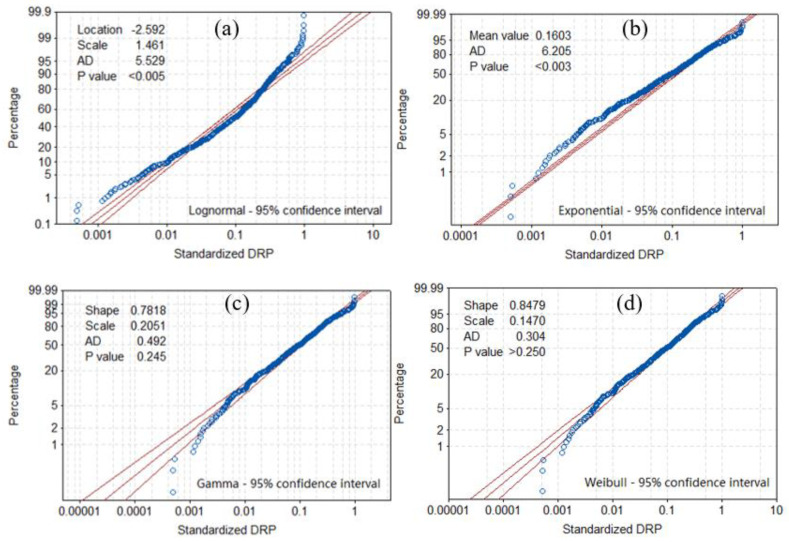
Probability plot of the DRP during the 2013 summer drought in the middle and lower reaches of the Yangtze River, China. (**a**) Lognormal > 95% confidence interval; (**b**) Exponential > 95% confidence interval; (**c**) Gamma > 95% confidence interval; (**d**) Weibull > 95% confidence interval.

**Figure 13 ijerph-17-07660-f013:**
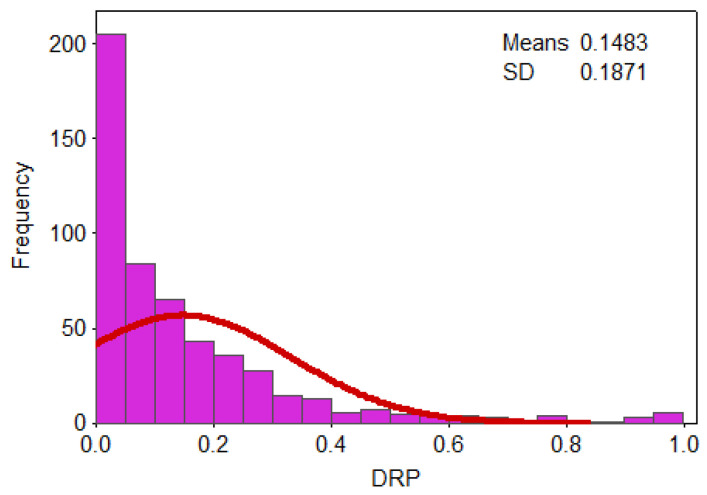
Probability density of the standardized DRP.

**Figure 14 ijerph-17-07660-f014:**
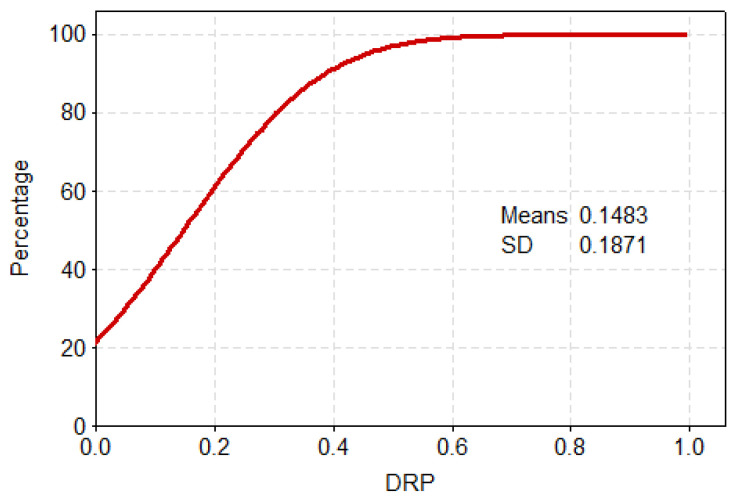
Cumulative probability of the standardized DRP.

**Figure 15 ijerph-17-07660-f015:**
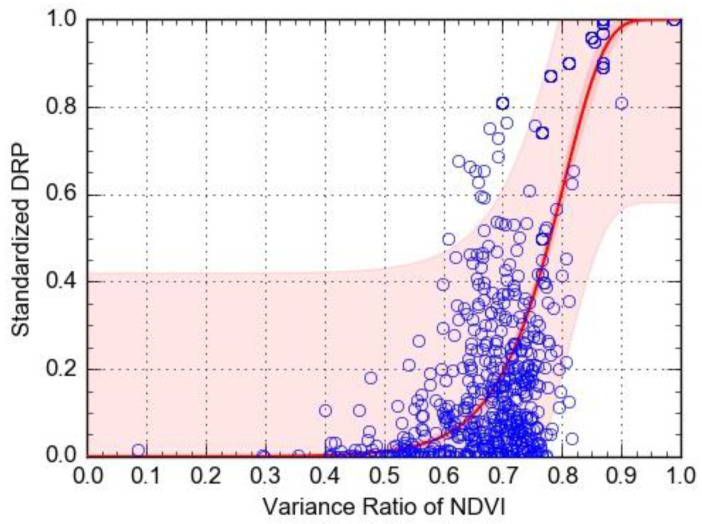
The vulnerability curve for the variance ratio of NDVI as a predictor of standardized DRP.

**Figure 16 ijerph-17-07660-f016:**
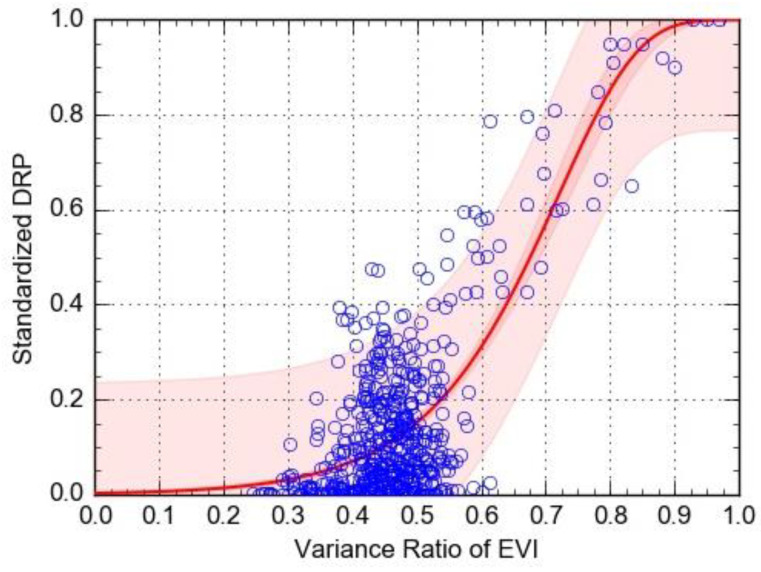
The vulnerability curve for the variance ratio of EVI as a predictor of standardized DRP.

**Table 1 ijerph-17-07660-t001:** Data sets used in this study.

Data Set	Content	Data Source	Year
Historical disaster data	Drought disaster data, at the county level, for the middle and lower reaches of the Yangtze River in 2013	National Disaster Reduction Center of the Ministry of Emergency Management of China	2013
Meteorological observation database	Precipitation, temperature, radiation, wind speed, relative humidity, and other daily data for all meteorological stations	China Meteorological Data Service, Center National Meteorological Information Center	1951–2013
Remote sensing indices data for drought	MODIS vegetation indices	https://ladsweb.modaps.eosdis.nasa.gov	July–September 2013
Basic geographic data	County administrative division boundaries, rivers, etc.	National Geomatics Center of China http://www.ngcc.cn/ngcc/html/1/index.html	2015

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
