# Peer review of "Vulnerability Analysis to Drought Based on Remote Sensing Indexes"

_ijerph, 2020, doi:10.3390/ijerph17207660_

Round 1
Reviewer 1 Report
Dear Authors,
Great improvement from the previous versión.
Best regards,
Author Response
Reply:
We appreciate for Reviewers’ warm review work earnestly. The comments are all valuable and very helpful for revising and improving our paper, as well as the important guiding significance to our researches. Once again, thank you very much for your comments and suggestions.

Reviewer 2 Report
Although the authors mention in the cover letter that they tried to improve the manuscript, I believe that no significant improvements were made based on the feedback from the 5 reviewers and therefore the manuscript, as it is, should not be accepted for publication.
Author Response
Reply:
The remote sensing technology is an important tool in this manuscript. Our authors focus on impact both for its relatively unusual nature (i.e., drinking water rather than agricultural loss) as well as its pragmatic purpose. We put interests in the application of commonly used remote sensing index products in drought vulnerability research. Our focus is on application. It is not particularly complicated in terms of remote sensing methods alone. Moreover, there are relatively few researches on the impact of drought risk on population by remote sensing. Now it is an exploratory research. Therefore, all the authors still believe that this manuscript is valuable and more suitable for IJERPH's special issue “Managing Disaster Risk in a Changing World”.

Reviewer 3 Report
This is my second review of your manuscript. I was reviewer 5 for the submission to Remote Sensing. The manuscript is improved on the whole. However, I believe equations 3 and 4 are in error, or I’m now confused by the variance ratio metrics description.
Assuming vegetation index values drop given drought, according to equations 3 and 4, the variance ratios should be negative. However, the legends for figures 4 and 5 range between 0 and 1?
The equations should the equations read NDVI1 (or EVI1) minus NDVIi (EVIi)?
Minor editorial correction:
Line 198 ‘urve’ should read ‘Curve’
Regarding Figures
Remote sensing would require that all map graphics include graticules. I’m not certain this applies to the IJERPH.
Author Response
This is my second review of your manuscript. I was reviewer 5 for the submission to Remote Sensing. The manuscript is improved on the whole. However, I believe equations 3 and 4 are in error, or I’m now confused by the variance ratio metrics description.
Assuming vegetation index values drop given drought, according to equations 3 and 4, the variance ratios should be negative. However, the legends for figures 4 and 5 range between 0 and 1? The equations should the equations read NDVI1 (or EVI1) minus NDVIi (EVIi)?
Reply:
Thank you for pointing this out. The normalized variance ratios were used in figures 4 and 5. It is true that the legends for figures 4 and 5 range between 0 and 1. Special thanks to you for your comments. It is really true as reviewer suggested that the equations should be NDVI1 (or EVI1) minus NDVIi (EVIi). We have made modifications according to the Reviewer’s comments (Page 5, line 179-182).
Minor editorial correction:
Line 198 ‘urve’ should read ‘Curve’
Reply:
We agree with this comment. Thank you for pointing this out. We have modified “Curve” according to the Reviewer’s comments (Page 7, line 203).
Regarding Figures
Remote sensing would require that all map graphics include graticules. I’m not certain this applies to the IJERPH.
Reply:
We hope that the current map graphics meet the requirements of the journal. Please inform us if we need to modify them.

Reviewer 4 Report
Confidential Comments to the EI:
This study with a target of DRP (drought at risk populations) , focused on the parameter estimation method was applied to determine the probability distribution of drought perturbations data by analyzing after the standardization of disaster data from the middle and lower reaches of the Yangtze River in 2013. The results of drought spatiotemporal pattern, drought disasters spatiotemporal analysis and vulnerability analysis were clearly presented. In addition, the results suggested that EVI (enhanced vegetation index) is more suitable to indicate drought vulnerability than NDVI (normalized vegetation index) for estimating the DRP by comparing the coefficient of determination (R2) values fitted for the vulnerability curve. However, a few points should be considered, especially for the discussion section. After the required revisions, I will recommend to publish.
Comments to the author:
Abstract: the key points are highlighted.
Introduction: this section is fine and highlight objectives of this study. However, some sentences need to be improved.
L42 “typically” changes “especially”
L45 “The research to date …” changes “ To date, the reaesrch…”
L56-57 Please rephrase this sentence.
L60-62 Please rephrase this sentence
L65-68 Please rephrase this sentence
Materials and methods:
This section is fine, the details of experimental setup and methods has been well described.
Results:
In this section, the main results have been described clearly, while there are some errors needed to be corrected.
L246 delete “both”
L248-250 Please rephrase this sentence
L307-310 Please rephrase this sentence, it may make readers confusing.
Discussion:
This section should be carefully improved. In this study, the major results were not enough discussed by citing more reference.
Author Response
Confidential Comments to the EI:
This study with a target of DRP (drought at risk populations) , focused on the parameter estimation method was applied to determine the probability distribution of drought perturbations data by analyzing after the standardization of disaster data from the middle and lower reaches of the Yangtze River in 2013. The results of drought spatiotemporal pattern, drought disasters spatiotemporal analysis and vulnerability analysis were clearly presented. In addition, the results suggested that EVI (enhanced vegetation index) is more suitable to indicate drought vulnerability than NDVI (normalized vegetation index) for estimating the DRP by comparing the coefficient of determination (R2) values fitted for the vulnerability curve. However, a few points should be considered, especially for the discussion section. After the required revisions, I will recommend to publish.
Comments to the author:
Abstract: the key points are highlighted.
Reply:
Special thanks to you for your good comments.
Introduction: this section is fine and highlight objectives of this study. However, some sentences need to be improved.
L42 “typically” changes “especially”
Reply:
We agree with this comment. Thank you for pointing this out. We have changed into “especially” in the revised manuscript (Page 1, line 42).
L45 “The research to date …” changes “ To date, the reaesrch…”
Reply:
We agree with this comment. We have changed into “To date, the research…” in the revised manuscript (Page 2, line 45).
L56-57 Please rephrase this sentence.
Reply:
We agree with this comment. We have rephrased this sentence in the revised manuscript (Page 2, line 57).
L60-62 Please rephrase this sentence
Reply:
We agree with this comment. Thank you for pointing this out. We have rephrased this sentence in the revised manuscript (Page 2, line 62).
L65-68 Please rephrase this sentence
Reply:
We agree with this comment. According to the Reviewer’s suggestion, we have rephrased this sentence in the revised manuscript (Page 2, line 67-70).
Materials and methods:
This section is fine, the details of experimental setup and methods has been well described.
Reply:
Special thanks to you for your good comments.
Results:
In this section, the main results have been described clearly, while there are some errors needed to be corrected.
L246 delete “both”
Reply:
Thank you for pointing this out. We have deleted “both” in the revised manuscript (Page 8, line 251).
L248-250 Please rephrase this sentence
Reply:
We agree with this comment. Thank you for pointing this out. We have rephrased this sentence in the revised manuscript (Page 8, line 253-257).
L307-310 Please rephrase this sentence, it may make readers confusing.
Reply:
We agree with this comment. Thank you for pointing this out. We have rephrased this sentence in the revised manuscript (Page 13, line 313-316).
Discussion:
This section should be carefully improved. In this study, the major results were not enough discussed by citing more reference.
Reply:
We agree with this comment. It is really true as reviewer pointed that the major results were not enough discussed. According to the Reviewer’s suggestion, we have added some discussions about the major results by citing more reference in the revised manuscript (Page 17, line 405-415).

Round 2
Reviewer 2 Report
Some improvements have been made to the manuscript and therefore I believe that it has enough quality to be published in the IJERPH.
This manuscript is a resubmission of an earlier submission. The following is a list of the peer review reports and author responses from that submission.
Round 1
Reviewer 1 Report
Dear Authors,
General comments:
The manuscript uses a method of assessing vulnerability to drought. Currently, studies related to climate change are a source of research and necessary to know its different effects, in this case to evaluate the scope and improve the prevention and action of the risk of drought. Great efforts by the authors!.
The content of the work is difficult to read and understand. It is recommended that the structure of the text be revised and the writing improved.
Please consider some comments and suggestions:
Title: The title of the manuscript is unclear as to the use of the term remote sensing
Abstract: Summarise the article's main findings and avoid repetitions or unnecessary information.
Keywords It is recommended to follow the "Instructions for Authors" for the keywords: “List three to ten pertinent keywords specific to the article; yet reasonably common within the subject discipline”.
Introduction: It is recommended to review the introduction section and provide sufficient background of similar investigations. Reference to similar previous work is very limited. There are many papers published during the last years.
Please refer to the rules described in the "Instructions for Authors" (https://www.mdpi.com/journal/remotesensing/instructions) for the images and maps, figure captions, citations, and file names meet their formatting requirements:
- Images and maps must have a scale, a north arrow and coordinates
It is recommended to follow the "Instructions for Authors" for the experimental and research data to be openly available either by uploading in a cloud file, or by publishing the data and files as supplementary information in this journal.
And comments more accurately:
Row 88: The acronym "DRP" has been described in the abstract section
Row 114: It is recommended to improve figure 1 and the attached information: label the Yangtze River, coordinates, etc.
Row 123-126: It is recommended to improve the writing: selection of the remote sensing index, characteristics of the downloaded scenes, etc.
Row: 133: It is recommended to mention the source of the "National Centre for Basic Geographic Information" (URL).
Row 135: There is a format error in the table.
Row 141: What do you mean by: “…the image was re-projected;…”?
Row 144 and 328: What is "MINTAB"? Mention the source.
Row 151-152: Improve writing and understanding.
Row 166: What do you mean by "venerable"?
Row 166: This expression is true: "Specifically, EVI improves on NDVI’s spatial resolution"? You can explain it?
Row 179: The value 2,5 of equation nº 2 is valid for the sensor MODIS?
Row 215: Are there other studies on the assessment of the drought vulnerability curve in areas or crops?
Row 245: There is a format error in the Figure 2. A caption on a single line should be centered.
Row 277: If there are multiple panels, they should be listed as: (a) Description of what is contained in the first panel; (b) Description of what is contained in the second panel. A caption on a single line should be centered.
Row 277: The index in Figures 7 and 8 will correspond to the EVI vegetation index
Row 285: Improve writing.
Row 288: It is recommended to mention the source of data
Row 336: If there are multiple panels, they should be listed as: (a) Description of what is contained in the first panel; (b) Description of what is contained in the second panel. A caption on a single line should be centered. Improve the explanation of the figures
Row 347: If there are multiple panels, they should be listed as: (a) Description of what is contained in the first panel; (b) Description of what is contained in the second panel. A caption on a single line should be centered.
I understand that the data analysed correspond to the year 2013 as there is a severe drought episode in the study area. Has a similar monitoring been carried out for the following years?
Kind regards,
Reviewer 2 Report
I found the subject of the manuscript interesting, but I believe that the manuscript might be improved by a more detailed description of the methods. Also, this will establish a clearer link between the methods and their corresponding results.
- In line 112, the province of Zhejiang is repeated in the text. Please add a grid with latitude and longitude to the right image in Figure 1 and improve its resolution.
- Add the location of the 90 meteorological stations (line 129) to Figure 1.
- Explain in more detail the method described in lines 138-140.
- The first author to propose NDVI (line 152) was not Deering (1978) but Rouse et al. (1973).
Rouse, J. W., Hass, R. H., Schell, J. A., & Deering, D. W. (1973). Monitoring vegetation systems in the great plains with ERTS. Third Earth Resources Technology Satellite (ERTS) Symposium, 1, 309–317. https://doi.org/citeulike-article-id:12009708
- Explain the methods described in lines 222-227 and link them to their corresponding results (section 3.3). Although you mention it in lines 412-419, further in the discussion section, it is still not clear.
- From the analysis of Figure 6, it seems that the cumulative probability of the 0.4-0.6 range is almost 20% rather than 12%, is this value correct? Again, from Figure 13 it seems that the cumulative probability of the 0~0.2 range is around 60% rather than 75%!
- In line 299, replace Figure 10 by Figure 9.
- In lines 313-318 you mention a lot of cities names that are not familiar to the reader. I suggest the addition of a Figure with the location of those cities differentiated by colour (300,000 and 600,000).
- Please explain how the coefficients of determination were obtained for the standardized DRP retrieved from NDVI and EVI. Were they compared with results from section 3.2.3? It is not clear!
Reviewer 3 Report
The article intends to analyze vulnerability to drought based on remote sensing indices.
Major issues.
Authors present the drought year 2013 as with the highest annual temperature reach since 1961, and one of the lowest yearly precipitation. Temperature and precipitation are two critical variables for drought, but not uniques factors. Authors assume that if rainfall was scarce and temperature high, vegetation was also under a severe drought year, which is not always true. It was also based on the 615 counties under drought declared by the National Disaster Reduction Center of the Ministry of Emergency Management. The authors should have used data (e.g., meteorological, remote sensing) to identify drought events, and the government data could be uses as validation. For example, have been used standardized precipitation index (SPI) or standardized vegetation index (e.g, zNDVI).
Later on, the authors present the remote sensing indices for drought. The NDVI and EVI are instead vegetation indices. The authors scaled the NDVI/EVI regard the start date. They used just data for 2013; to evaluate drought, you need to use historical data (> 20 years) as drought is a climatic disaster. The spatial variability showed by the vegetation indices could be due to different species, stage of development, bare soil, probably drought. But, you cannot compare the heterogeneous type of vegetation as they were all the same, is biased. To a more fair analyzed there have been proposed vegetation based drought indices as the zNDVI (Peters et al., 2002), VCI (Kogan 1995), zcNDVI (Zambrano et al., 2018)
Peters, A. J., Walter-Shea, E. A., Ji, L., Viña, A., Hayes, M., & Svoboda, M. D. (2002). Drought monitoring with NDVI-based Standardized Vegetation Index. Photogrammetric Engineering and Remote Sensing, 68(1), 71–75.
Kogan, F. N. (1995). Application of vegetation index and brightness temperature for drought detection. Advances in Space Research, 15(11), 91–100. https://doi.org/10.1016/0273-1177(95)00079-T
Zambrano, F., Vrieling, A., Nelson, A., Meroni, M., & Tadesse, T. (2018). Prediction of drought-induced reduction of agricultural productivity in Chile from MODIS, rainfall estimates, and climate oscillation indices. Remote Sensing of Environment, 219, 15–30. https://doi.org/10.1016/j.rse.2018.10.006
The rest of the study is based on this vegetation indices assumed as drought indices. For that reason, I will not recommend the article for publication in the Remote Sensing journal.
Reviewer 4 Report
The manuscript has used two remote sensing indices to assess vulnerability of drought in 2013 using drought at risk population (DRP). The drought vulnerability curve has been developed from NDVI and EVI indices. The manuscript is characterizing drought vulnerability but somehow the methodology is not convincing. For instance, all drought related prior studies have employed various drought indices which integrated vegetation indices (NDVI and EVI) along with the surface temperature and evapotranspiration information (TCI and VTCI).
However, in this study, the vulnerability curve is simply based on NDVI and EVI which doesn’t explicitly provide any information related to environmental stress. The method of developing DPR is not clear in the methodology section but appeared lately under result section. It needs restructuring the manuscript. The figures 3 and 4 related to NDVI and EVI are simply remote sensing data but presented as results. It was misrepresented and used for describing drought conditions. If simple NDVI and EVI can capture accurate drought conditions, then why so many studies are conducted on VTCI, TCI, VCI, SPI, PDSI, and so on. Overall, this manuscript has methodological flaws which need substantial restructuring of the manuscript.
Reviewer 5 Report
The authors investigated the relationships among drought metrics as derived from remote sensed vegetation indices and a metric of impact, namely drought at risk population to evaluate whether an established link can be used for rapid drought assessment and mitigation planning.
I appreciate this focus on impact both for its relatively unusual nature (i.e., drinking water rather than agricultural loss) as well as its pragmatic purpose. And I think it stimulating to include an application of this nature in Remote Sensing. However, I think additional information is needed within the text. Some of the metrics used (variance of ratio) are not explained and should be for Remote Sensing readers. And the discussion, which currently reads solely as a defense against the argument that the period of study was too limited, should be expanded to consider additional potential pitfalls of the approach and consideration of the remote sensing aspect of the analysis. More detailed comments to this effect are next provided, before simple grammatical changes and improvements to figures are suggested.
Comments
Comments
215 – 218 This discussion of “body” is not very clear/confusing. What is meant by ‘body’?
The variance ratio is a useful measure in this context, but there is absolutely no discussion of it in the methods section. Rather, it is first introduced in the legend of a figure. Worse still, the color ramp used for figure 5 is similar to those often sued to portray NDVI or EVI values themselves, not their variance ratios. This is very confusing for the reader. Please describe this metric in the methods section, and clarify that it has been applied to each pixel as calculated from the 24 time steps (images) available during the drought period of 2013 (correct?).
Why are the sample sizes different between NDVI (745) and EVI (533) and does the cause of this difference explain why EVI would be more useful than NDVI?
I’m not convinced a difference of 2% in explained variance supports the assertion that EVI is a superior metric to NDVI for this application. There are other aspects of vegetation indices that make NDVI more commonly employed compared to EVI – that might make it more generally appropriate for such an analysis. I don’t know this is the case. I’m simply pointing out that a 2% difference for a single study year in a single area of the world (albeit a large one) should yield a conclusion that EVI is more suitable for this purpose.
Other suggestions related to readability
166 The same benefits of EVI relative to NDVI are stated twice – the second version being more informative. I suggest you remove the second sentence on line 166 and move the related references to the following statement.
188 The sentence starting with “It is…” is essentially a repeat of the preceding sentence. I suggest you remove it and assign the citations to the previous sentence.
207 Delete “, however”
210 “disaster rate curve needs a close to the quotation: “disaster rate curve”
215 – 218 This discussion of “body” is not very clear/confusing. What is meant by ‘body’?
252 change “2013; these” to “2013. These”
281-286 replace all “;” with periods and capitalize the first word of each sentence following the period.
Figures
While the geographic coordinates (actually the ranges in latitude and longitude) are discussed in the description of the study area, all map figures should include graticule (lat/lon or projected coordinate values) information as per journal requirements.
How were the meteorological data summarized? Were they simply averaged across all stations for the purpose of generating these figures?
Tables
Put sample sizes in the table. Obviously, the meteorological data was available for longer than 2013, although that is all that is indicated in the table.